# Acceptability of the voice your values, an advance care planning intervention in persons living with mild dementia using videoconferencing technology

**Shirin Vellani**[1]*, **Martine Puts**[2], **Andrea Iaboni**[3,4], **Katherine S. McGilton**[2,3]

**1** Faculty of Health Sciences, School of Nursing, McMaster University, Hamilton, Ontario, Canada, **2** Lawrence S. Bloomberg Faculty of Nursing, University of Toronto, Toronto, Ontario, Canada, **3** KITE, Toronto Rehabilitation Institute, University Health Network, Toronto, Ontario, Canada, **4** Department of Psychiatry, University of Toronto, Toronto, Ontario, Canada

* shirin.vellani@mail.utoronto.ca

**Data Availability Statement:** Data is well incorporated in the manuscript. Full data contain personally-identifying information, such as potentially identifying information, such as

## Abstract

Advance care planning (ACP) can improve outcomes for persons living with dementia (PLwD). Clinicians see the lack of acceptability of these conversations as a barrier to ACP in individuals with mild dementia. COVID-19 pandemic has magnified the need for ACP discussions in older adults, particularly for those living with dementia. In light of the pandemic, much of the healthcare is provided virtually, but little evidence exists on how to best implement ACP virtually. We designed Voice Your Values (VYV), a tailored ACP intervention for persons living with mild dementia and their trusted individuals such as friends or family. **Purpose** Determine the acceptability of the VYV intervention, in terms of its content and the potential utility of videoconferencing to deliver it. **Methods** For this pilot study, we recruited 21 dyads of older adults with mild dementia and their trusted individuals from five geriatric clinics in Ontario, Canada. The tailored VYV intervention was delivered to dyads over two sessions over videoconferencing. Acceptability was assessed using scores on a modified Treatment Evaluation Inventory. The interventionist diary and Researcher Virtual Experience Questionnaire were used to examine facilitators and barriers, whereas Participant Virtual Experience Questionnaire was used to understand their experience. Qualitative data was analyzed using inductive content analysis. **Results** 100% of the participants rated VYV as acceptable. Participants and researcher rated video and sound quality highly. PLwD who lived with their trusted individuals were more likely to find the intervention acceptable ($t$ = 3.559, $p$ = 0.001, β = 0.323). Five interrelated themes were established that describe the acceptability of the virtually delivered VYV intervention. All PLwD were able to articulate their values and wishes related to being in a terminal and vegetative states and had them documented. **Conclusion** The virtual VYV intervention was an acceptable approach to ACP in older adults with mild dementia and their trusted individuals.

recruitment site information, and therefore are subject to ethical and legal restrictions on publicly sharing. Therefore, we cannot share the data set. Specific questions could be shared with the corresponding author and they will address any reasonable requests." We confirm that our manuscript includes the "minimal data set" as defined by the PLOS.

**Funding:** The author(s) received no specific funding for this work.

**Competing interests:** The authors have declared that no competing interests exist.

# Introduction

Individuals living with dementia progressively lose their cognitive capacity to participate in their own care decisions when in the advanced stages and at the end-of-life [1]. Subsequently, trusted individuals, such as family and friends of persons with dementia, are expected to make critical care decisions and often find it challenging [2, 3]. The process of advance care planning (ACP) allows an adult to think about their values and wishes regarding future care, and share these with those they trust, in order to receive care that is consistent with what they expressed [4].

ACP is recommended for individuals living with dementia, yet it is uncommon in clinical practice [5–7]. It is estimated that less than one-third of persons living with dementia (PLwD) have any ACP discussions with clinicians in specialized geriatric or primary care practices [8]. As a result, those living with dementia often experience inadequate end-of-life care [2, 9]. Numerous factors have been identified by clinicians as barriers to initiating ACP conversations in PLwD including unpredictable illness trajectory; uncertainty about the right time to commence the ACP process [10–12]; lack of time, resources, and knowledge; discomfort with ACP discussions [8, 13]; and difficulty in determining the decision-making capacity of the PLwD [14].

Findings from previous studies also highlight concerns about the acceptability of ACP discussions in persons with dementia and their care partners. For example, Fried and colleagues reported that many patients with early stages of cognitive impairment and their care partners are not interested in ACP conversations as they want to live in the moment, think dementia is unrelated to physical health, or do not see the point in partaking in ACP as they would not be aware of the decisions made for them in the advanced stages due to cognitive decline [15]. Clinicians also suspect that PLwD and their trusted individuals may not engage in ACP discussions as they may be in denial or uncomfortable discussing death [8], and may feel overwhelmed with the ACP process [16]. As such, there are barriers related to concerns about the acceptability of ACP discussions in PLwD and their trusted individuals.

There remains a need to create more opportunities to engage PLwD and their trusted individuals in the ACP process. Therefore, we designed Voice Your Values (VYV), a tailored ACP intervention. The VYV intervention was initially developed for face-to-face delivery in a clinic or PLwD's homes, but it was not tested in this format due to the physical distancing measures instituted in light of the COVID-19 pandemic. It was therefore adapted to be delivered virtually using principles of remote methods for dementia research, as prescribed in a practical guide we developed with researchers who are experts in the field [17]. Despite the widespread use of virtual care during the pandemic, there is no specific evidence to guide the practice of virtual ACP. To our knowledge, this is the first study that engaged older adults with mild dementia and their trusted individuals in the ACP process during the COVID-19 pandemic through an online video conferencing platform. The study explored the following research questions:

1. What is the acceptability of the VYV intervention as perceived by the participants in terms of its content and activities?

2. What factors influence the acceptability of the VYV intervention in participants?

3. What are the participants' and researcher's experiences with videoconferencing for the VYV intervention?

## Methods

### Design

A pilot study was conducted to determine the feasibility, preliminary efficacy, and acceptability of the VYV intervention. The data related to the feasibility (recruitment, retention, and intervention fidelity) and preliminary efficacy are described elsewhere [18]. The findings of the feasibility of the VYV intervention are promising, and the outcomes of preliminary efficacy demonstrated improvement older adults as well as trusted individuals. The current paper reports on the acceptability and perceptions of participants related to the intervention. Ethics approval was received from the University Health Network (UHN) Research Ethics Board, University of Toronto, as well as the recruitment sites.

### Participants

Community-dwelling older adults were eligible if they were 65 years or older, had mild dementia as diagnosed by a nurse practitioner or physician, were able to read and speak English, had access to a device with internet for videoconferencing, and had a trusted individual who had at least weekly contact with the PLwD and could enroll as a dyad. PLwD with acute psychotic disorders and/or clinical depression, mild cognitive impairment, blindness, or deafness, or those who already had a written ACP (i.e., a written document identifying their values, wishes, and care goals for future) in place were not eligible. Eligibility criteria for the trusted individuals included: being 18 years of age or older; no self-reported dementia, blindness, or deafness; access to a device for videoconferencing; and ability to read and speak English. The VYV intervention was delivered to participants by an interventionist (PI) who is a trained nurse practitioner with extensive experience working with older adults in a variety of specialized geriatric settings.

### Consent

The VYV study was completed as a research project where participants were recruited from five geriatric clinics located in geographically diverse regions of Ontario, Canada. Recruitment and data collection lasted from July 2020 to February 2021. Potential participants were identified by the most responsible provider (physician or nurse practitioner) at the recruitment sites, who shared their names with the PI (SV) after seeking their verbal consent for it. Consent procedure was conducted by the PI over the Health Information Protection Act, 2016 compliant, Microsoft (MS) Teams videoconferencing platform. Enhanced consenting techniques were used [19] and participants' comprehension was assessed using the teach-back method [19]. Verbal consent was obtained from participants (PLwD and trusted individuals), where each one of them served as a third party witness for the other. Participants were sent the PI signed consent form for their record, while its copy and the audio recording of the consent procedure were uploaded on the UHN server as records.

### Voice your values intervention

The VYV intervention was designed based on empirical data and guided by the theoretical underpinnings of the Representational Approach to Patient Education [20–23] and the Transtheoretical Model of Stages of Change [24]. The process of delivering the intervention was informed by the recommendations developed by Piers et al. for ACP in PLwD [25]. VYV was offered over two sessions to each PLwD and trusted individual dyad in the PI. The NP interventionist did not work in any of the five participating geriatric clinics. The first VYV session offered participants an opportunity to share their perspectives on living with dementia,

including fears about the future when dementia progresses, their understanding of ACP in the context of dementia, as well as what it means for the PLwD to have a good life and what might they consider unacceptable. The second VYV session involved the provision of tailored education to the dyad based on the information acquired in the first session, and coaching the PLwD to think about, and share their values and wishes for future care related to being in a terminal and/or vegetative state. A document of expressed values and wishes was created for the dyad for further discussion with clinicians and others, and as a resource for the trusted individuals to make informed care decisions in the future.

## Measures

Data was collected virtually over MS Teams platform from all participants 1–2 weeks post intervention session two by a Research Assistant (RA), a registered nurse hired for the study and trained on the measures and methods of virtual data collection. The RA had no prior relationship with any of the participants. During the data collection, the RA displayed the responses for each item/question of the measures using the screen sharing option for the ease of selection. While collecting data, the RA spoke slowly and repeated questions as necessary, making sure questions were adequately comprehended. Data was acquired from each participant of the dyad separately to prevent the influence of the other's presence on responses. These sessions were video recorded so the RA could provide undivided attention to the participants during the data collection. Immediately after the session, RA inputted all the data electronically directly on the UHN drive by viewing the recordings. All participants were asked about their sociodemographic information pre-intervention. PLwD's past medical history and last cognitive test scores were acquired from the clinics after seeking their consent.

**Acceptability.** To measure the acceptability of VYV, the Treatment Evaluation Inventory (TEI) was administered to the PLwD and their trusted individuals [26]. It has been used in several studies involving older adults, their healthcare staff, and families for the assessment of treatment acceptability [26–28]. It tests the overall reaction to the treatment focusing on acceptability, perceived effectiveness, as well as risks and accompanying side effects [26]; and consists of 11 items on a 7-point scale with a total score ranging from 11 to 77. The VYV intervention was to be considered acceptable if >75% of the respondents scored $\geq$ 47 (75%) on the modified TEI acceptability questionnaire, based on a previous ACP study in persons with end-stage renal disease [29]. For the current study, TEI was adapted to specify the treatment as the VYV intervention, one of the 11 questions related to improvement in symptoms was removed, and six supplementary open-ended questions were added, resulting in a total of 16 questions. Internal consistency was moderate in the current study, as indicated by the Cronbach alpha coefficients of 0.74 for the TEI global scale, 0.67 for the general acceptability subscale that consisted of 7 items, and 0.65 for the negative aspects subscale consisting of 3 items.

**Experience with virtual modality.** To acquire feedback from participants on the suitability of videoconferencing for the delivery of the VYV intervention, a Virtual Experience Questionnaire-Participants (VEQ-P) was administered to all participants [30]. This questionnaire has 3 Likert-type questions on the technical aspects of video calls and 3 open-ended questions assessing participants' perspectives on the use of videoconferencing for conducting this research. Specific qualities of sound and video were rated using a 5-point scale on the ability to see and hear where 1 represented none of the time and 5 represented all of the time; whereas frequency of lags in video were captured as: 1 meaning, none of the time to 5 meaning all of the time.

A Virtual Experience Questionnaire-Researcher (VEQ-R) version was completed after each intervention session by the interventionist to rate the experience with technology, and

document observations regarding the dyad's level of comfort using a real-time virtual platform. These questionnaires were adapted from a previous study that explored the suitability of Zoom for qualitative data collection [30]. In addition, PI completed a diary after each session that recorded time to complete each session, specific activities carried out, dynamics between the participants, barriers and facilitators; participants' opinions and perspectives towards VYV, topics of education and any other observations that could help refine the intervention in future trials. This data was then used for triangulation with participants' qualitative data and helped with intervention fidelity described elsewhere.

### Analysis

Statistical analyses were conducted with the SPSS IBM Statistical Software version 27.0 with an α of 0.05 to analyze descriptive statistics. Standard linear regression was used to explore the influence of select predictors on the acceptability scores in PLwD and trusted individuals separately. Predictors entered in the regression equation included sex of the participants, time in days since dementia diagnosis, and whether the trusted individual lived with the PLwD, assuming that daily interaction with the PLwD may have a positive impact on the trusted individual's level of acceptability of ACP conversations.

Qualitative information from the interventionist's diary logs, the acceptability questionnaire, as well as the VEQ-P and VEQ-R were analyzed using the inductive approach for Qualitative Content Analysis [31, 32]. The data was first coded independently by the PI (SV), and then corroborated with the Senior Author (KM). Codes were then grouped into main and subcategories.

### Rigor

To ensure quality, rigor, transparency, and completeness, the CONSERVE (CONSORT and SPIRIT Extension for RCTs Revised in Extenuating Circumstances) checklist was used [33]. We also used the template for intervention description and replication (TIDieR) checklist to describe details of the VYV intervention for future replication [34]. An audit trail was created by the PI to maintain credibility and rigour throughout the data analysis process, and to allow for the review and examination of each step [31]. Each step of the coding process was reviewed with the senior author (KM), and feedback was incorporated in devising the final categories, subcategories, and their descriptions. Final categories and subcategories were reviewed and agreed upon by the full research team. NVivo 12 was used for the management of the qualitative data.

### Results

The sample included 21 dyads of older adults living with mild dementia and their trusted individuals. There was an equal number of male and female PLwD with a mean age of 80 ± 6.6 years. The trusted individuals' mean age was 62 ± 11 years; majority were children (n = 11, 55%), with 35% (n = 7) being daughters and 20% (n = 4) being sons; the rest were spouses (n = 9, 45%). Most of the dyads were white (n = 11, 55%), followed by South Asian (n = 5, 25%). Table 1 presents the characteristics of the participants.

### Quantitative findings

This section reports on the quantitative data related to the acceptability of the VYV intervention (research question 1), factors influencing the acceptability (research question 2) and the virtual experiences of participants and researcher using the TEI, VEQ-P and VEQ-R

**Table 1. Characteristics of participants.**

| | PLwD | Trusted Individuals |
|---|---|---|
| | **(n = 20)** | **(n = 20)** |
| Quick Dementia Rating Scale Scores, Mean (SD) | 8.3 (3.3) | |
| Days since diagnosis, Mean (SD), | 660 (439) | |
| Range | 92–1450 | |
| **Common Chronic Conditions with Dementia, n (%)**** | | |
| • Musculoskeletal conditions | 12 (60) | |
| • Hypertension | 11 (55) | |
| • Arrhythmias | 7 (35) | |
| • Coronary artery disease | 6 (30) | |
| • Cancers | 5 (25) | |
| • Chronic kidney disease* | 4 (20) | |
| Age, Mean (SD), | 80 (7) | 62 (11.5) |
| Range | 67–91 | 44–81 |
| **Sex, % Females** | 50 | 65 |
| **TI's Relationship, n (%)** | | |
| • Spouse | 9 (45) | |
| • Daughter | 7 (35) | |
| • Son | 4 (20) | |
| **Relationship status n (%)** | | |
| • Married | 11 (55) | 14(70) |
| • Widow/widower | 5 (25) | 0 |
| • Separated/divorced | 3 (15) | 3 (15) |
| • Single | 1 (5) | 3 (15) |
| **Education n (%)** | | |
| • High school or less | 11 (55) | 1 (5) |
| • Non-university training | 5 (20) | 7 (35) |
| • College/university degree | 3 (15) | 12 (60) |
| **Race/ethnicity n (%)** | | |
| • Black | 2 (10) | 2 (10) |
| • White | 11 (55) | 11 (55) |
| • Chinese Asian | 1 (5) | 1 (5) |
| • South Asian | 5 (25) | 5 (25) |
| • West Indian | 1 (5) | 1 (5) |
| **Employment status n (%)** | | |
| • Full time | | 10 (50) |
| • Currently unemployed | | 1 (5) |
| • Retired | 20 (100) | 9 (45) |

*1 person on hemodialysis

**missing for 1 PLwD.

questionnaires (research question 3). Of the twenty one PLwD, 18 completed all study procedures. One PLwD decline to participate next day after completing the consent, one declined to answer the TEI questions due to tiredness and was not willing to reschedule to complete the data collection. Another PLwD missed two appointments for the outcome data collection as they went for a walk and could not be located by their trusted individual at the time, missing the two-week window for data collection. All participants deemed VYV as acceptable, with

**Table 2. Researcher and participants' experiences of virtual modality.**

| Quality | Researcher Rating (n = 1) 40 sessions | PLwD Ratings (n = 19) | Trusted Individual Ratings (n = 20) |
|---|---|---|---|
| | Average (SD) | Average (SD) | Average (SD) |
| Sound | 4.8 (.75) | 4.9 (.23) | 4.5 (.76) |
| Video | 4.7 (.45) | 4.8 (.37) | 4.7 (.48) |
| Video Lag | 1.3 (.50) | 1.3 (.56) | 1.7 (.63) |

PLwD scoring an average (SD) of 54.7(8.9) and trusted individuals scoring an average of 58.9 (4) on the TEI acceptability questionnaire. There were no differences in the virtual experiences of PLwD and trusted individuals according to their responses on the VEQ-Participants questionnaire (See Table 2). Overall, the participants and the researcher rated the video and sound quality highly, and experienced minimum lags in video (See Table 3). Regression analysis revealed that older adults who lived with their trusted individuals were more likely to find the intervention acceptable compared to those who didn't live with their trusted individuals ($t$ = 3.559, $p$ = 0.001, $\beta$ = 0.323). Participants' sex and time since dementia diagnosis did not have any impact on the TEI acceptability scores in PLwD or trusted individuals (Table 3).

## Qualitative findings

In addition to the above quantitative analysis, the qualitative data from the TEI acceptability, VEQ-P, VEQ-R questionnaires, and the interventionist diary log provided further insight into the acceptability of the VYV intervention (research questions 1 and 3). Qualitative findings were organized into two broad categories: 1) acceptability of the VYV intervention; and 2) researcher and participants' experiences with videoconferencing. See Table 4 for these categories, respective subcategories, and their description. S1 Table presents additional illustrative quotes for each subcategory.

**1) Acceptability of the VYV intervention.**   While there were some differences between PLwD and trusted individuals in terms of the acceptability of the VYV intervention regarding its content and activities, similar themes were identified between them and thus a combined data analysis was conducted. Five interrelated categories were established that describe the

**Table 3. Standard multiple regression results for TEI acceptability scores.**

| | Participants | |
|---|---|---|
| Predictors | PLwD | Trusted Individuals |
| | (n = 18) | (n = 20) |
| Time since dementia diagnosed in PLwD, days | -.002 (-.005 - .002) | .000 (-.005 - .005) |
| $\beta$ (95% CI) | | |
| Participants' sex, Female | .991 (-2.173–4.154) | -.440 (-4.689–3.809) |
| $\beta$ (95% CI) | | |
| Trusted individual lives with PLwD | 6.309* (2.797–9.82) | .668 (-3.979–5.316) |
| $\beta$ (95% CI) | | |
| | Adjusted $R^2$ | |
| | .085*[a] | |
| | -.178[b] | |

* $p$ = .001; CI Confidence Interval; PLwD Person living with dementia

[a] PLwD

[b] Trusted Individual.

**Table 4. Categories and subcategories related to the acceptability of experiences with VYV.**

| Categories | Subcategories | Description of Subcategories |
|---|---|---|
| 1- Acceptability of the VYV intervention | a- Breaking the ice<br>b- Revealing the values and wishes for future care<br>c- Getting all the cards on the table<br>d- Coming to grips<br>e- A third person to bounce it off | • VYV as a means to engage in ACP process which the participants had not done before<br>• VYV as a medium for PLwD* to express values and wishes and for trusted individuals to learn about them<br>• VYV as an opportunity for dialogue among participants and ask each other questions<br>• VYV as a tool for the trusted individuals to prepare for future decision-making<br>• Value in presence of knowledgeable facilitator/clinician |
| 2- Researcher and participants' experiences with videoconferencing | a- Opinions about videoconferencing<br>b- Challenges with videoconferencing | |

*PLwD Person living with dementia.

acceptability of the virtually delivered VYV intervention. These include: a) breaking the ice, b) revealing the values and wishes for future care, c) getting all the cards on the table, d) coming to grips, and e) a third person to bounce it off. Details on the categories are presented below. The above categories were conceived as describing the acceptability of the intervention as they demystify some of the myths deterring clinicians to initiate ACP conversations in PLwD. Details on each subcategory are presented below.

*a. Breaking the ice.* This subcategory describes that for both PLwD and trusted individuals, VYV served as a means to engage in the ACP process that they had not had the opportunity to participate in before. They stated that these discussions helped them learn about what to expect with the progression of dementia, as well as how to prepare for it and create space for discussions between them. Several PLwD mentioned that they did not think it was important to express their values and wishes for future care because they may not be aware of their surroundings then, and they trust their care partners. Many believed that their trusted individuals are already aware of their values and wishes while trusted individuals denied having clarity on this and some even shook their heads in disagreement. However, participation in the VYV study made them realize that verbalizing what matters to PLwD will aid care partners in making difficult decisions and choices. The PLwD also acknowledged that making their wishes known can impact their quality of life when they are in the advanced stage of dementia and/or nearing the end-of-life. One PLwD explained:

> Well, I know now that it is important, and I didn't realize it would be that important. I thought that my caregiver could make the decisions for me and that would be fine. So, that was the biggest thing I got out of it. And it's maybe good to have wishes and values. It's the disease I have in my head, you're not really with it in the end anyways. So, them (care partners) knowing my values can help them to make decisions. (PLwD 10)

Trusted individuals expressed a sense of appreciation for having learned the views of their PLwD. Several participants stated that they had a sense of apprehension about the impending loss of their PLwD's decision-making capacity and their own ability to make difficult care decisions. Many expressed that VYV allowed PLwD to openly discuss these topics that they had previously deflected or were "evasive" about. One trusted individual explained:

> We have come pretty far, mom is not a person that opens up enough so it's always hard for us to figure out exactly what she wants. She always goes around the issue and wouldn't answer

*anything directly, so it was nice that you were able to coach this much from her. . . the fact that we got mom to let us know what her wishes are so that's what the main thing for me and I am happy with that. (TI 11)*

Several participants expressed that ACP discussions should be part of dementia care. Being involved in VYV where the facilitator discussed topics not mentioned before was seen positively and created opportunity for further discussion. The study allowed all participants to engage in the ACP process, either through thinking and sharing their values and wishes, or learning about them if they were trusted individuals.

*b. Revealing the values and wishes for future care.* The second subcategory demonstrated that VYV served as a channel for the PLwD to relay to the trusted individuals, their values and wishes for care while they are able to speak for themselves. Many PLwD shared that if they are unable to interact with others and their environment, they would rather die. Several conveyed the desire to pursue medical assistance in dying (MAID) when they lose their cognitive abilities. Some participants were interested in learning about advocating for changes in laws to allow for advance consent for those living with dementia. One PLwD who receives hemodialysis three times every week for end-stage renal disease shared:

> *If I get wacko, if this (referring to dementia) gets further worse and worse so I don't even know who I'm talking to or anything, then I would hope that they would put me in hospice and let me go, take me off of dialysis. (PLwD 19)*

Several PLwD shared their experience of witnessing close friends and family members who died of dementia, which served as an impetus for them to verbalize at what point care and life quality would be unacceptable for them when they reach the advanced stage of dementia. Many expressed mixed emotions when discussing the decision to end suffering and receive life sustaining treatments. On the one hand, they did not want to suffer at the end-of-life and receive any life sustaining treatments. On the other, they were sad for the grief their family would experience due to their passing. As well, many appreciated that being in a long-term care home or hospital would be a source of stress for their care partners.

> *I don't want to be attached to any tubes for any length of time, I don't think it's going to make any difference in the end, if there is no value in life that I can have, then why put on something that just keeps me alive, I think they (family) should respect my wishes. (PLwD 16)*

> *When life no longer holds any joy, and I can no longer look outside the window and enjoy things like snowfall, deer and birds, going out to smell fresh air, being with my grandkids, I would want my husband to let me go. (PLwD 15)*

Inevitably, the VYV intervention provided PLwD an opportunity to think about their wishes and share them, in turn increasing awareness of their trusted individuals about the type of care they desire. This trusted individual expressed:

> *It (referring to VYV discussions) just changes a lot of things in terms of the relationship and the future. I think it pinpoints some areas I need to take better care. I think one thing that really stands out is the question about if she was put on life support. It is something that we never discussed, so by her response I learned a lot about what she wanted. I think that particular question was the stand out for me on this. (TI 5)*

*c. Getting all the cards on the table.* This subcategory demonstrates that participating in VYV provided an opportunity for dialogue to PLwD and their trusted individuals, to ask each other questions and clarify their own thoughts about future care when the PLwD is terminally ill or in a vegetative state and unable to speak for themselves. For example, one PLwD (# 15) stated that she would want her family to respect her wishes that she expressed during the VYV sessions, saying that *"it's my life, when it's time to go, follow my wishes kids and let me go"*. Her husband, who attended the sessions as her trusted individual, said *"I am listening and I am taking it in (long pause), and I agree with her"*. The PLwD expressed that her husband has always been very religious, and she is worried that his beliefs would impact his decision-making for her end-of-life. Through the VYV sessions, they got to discuss her wishes openly and she felt relieved, while also appreciating that *"this must be very difficult for him (referring to husband)"*.

When talking about life sustaining treatments, one PLwD (# 14) asserted, *"all you are doing is prolonging the agony and taking up a bed in the hospital; I am very pragmatic"*. Hearing this comment, his wife (trusted individual) remarked, "*he should at least get water, such as through an IV*". The PLwD immediately interjected, *"I don't think so"*. Hence, we discussed the goals of care in this late stage. They both agreed that at the point when there is no coming back, and he is unable to swallow, he should be kept comfortable but without tubes and machines. The PLwD articulated, "*we don't let the dogs suffer, why we do it to humans.*" The VYV sessions played a critical role in allowing PLwD to reflect and openly share their thoughts, while the trusted individuals got the opportunity to clarify what they heard.

*d. Coming to grips.* This subcategory is characterized by trusted individuals acknowledging the impending loss of PLwD's decision-making and communication abilities, and their own role in future decision-making. One trusted individual explained:

> It forced us to think again that he really has Alzheimer's, and some time I will need help and that he may not be able to make his decisions, and I will be making his decisions, so I need to know. If he has made these decisions, that means I don't have to, because that is a real burden for a lot of people. I think this (talking about VYV) is relieving the burden on the caregivers. I think caregivers have enough burdens without having to think how their loved ones want to be treated when they can't say it. (TI 16)

Many trusted individuals expressed that they avoided ACP conversations due to lack of knowledge on how to broach it with the PLwD, fear of causing emotional distress, and/or believing that the dementia diagnosis had diminished the PLwD's ability to engage in these discussions. They stated that VYV opened the door for further discussions with their PLwD on planning care and identifying contingencies in case the trusted individual gets sick or dies before the PLwD. Several trusted individuals also expressed feeling a decrease in stress and potential "guilt" and "agony" associated with future decision-making as a result of learning about their PLwD's wishes, particularly in the event that they require life sustaining treatments as part of care.

> It is good for me to hear my mother answer some of those questions. So, I think it helps, it eases the stress of being a caregiver, and it gives my mother a voice which is really important. (TI 7)

*e. A third person to bounce it off.* The final subcategory relates to the impact of the presence of a knowledgeable clinician to facilitate ACP conversations. Examples and discussions on this came from both the PLwD and their trusted individuals who shared their perspectives on the

positive influence of the presence of the interventionist trained in the care of older adults. One PLwD explained:

*SV (PI) was able to extract from me and fine tune my values. I had a general idea, but she was able to come up with examples to help me express what I feel. Better, I hope, than what I would have done trying to do this on my own. I cannot imagine trying to do this on my own. This has just been a God sent. (PLwD 15)*

Several participants indicated that due to the interventionist's training and experience, she used communication approaches that were easy to comprehend for the PLwD and led them to share their thoughts. For the trusted individuals, the interventionist's presence allowed them to learn about dementia, its stages, person-centered communication strategies, and designing a plan to prepare for future care and decision-making for their PLwD, as well as caring for their own wellbeing. Many expressed that this type of intervention should be offered through their geriatrician's office or the local Alzheimer's society. Trusted individuals pointed out that having ACP conversations in a "hypothetical way" in advance is much more preferrable to thinking about life sustaining treatments in a crisis situation. One trusted individual described:

*If I was to bring it up to her (referring to wife/ PLwD) without this intervention, then I wouldn't know where to begin; it could possibly go very hurtful and very destructive to a marriage. I think there is a need for somebody like SV or through Alzheimer's society to negotiate this. I think this might be one of the best discussions about Alzheimer's that we have had to date. (TI 15)*

**2. Researcher and participants' experiences with videoconferencing.** *a. Opinions about videoconferencing.* Participants' opinions on videoconferencing were predominantly positive. Many appreciated that it saved them travel time and costs. They also appreciated being at home due to the COVID-19 pandemic. Of the 40 participants (PLwD and trusted individuals), only one trusted individual indicated a preference for telephone over videoconferencing as it required her to travel to another city, where her father lived, to set up the technology. The participants felt that videoconferencing was the closest substitute for in-person meetings for ACP conversations, as they could observe each other's expressions. One trusted individual said:

*Definitely not the telephone, if the in-person was available yes, that would be nice but seeing someone on the screen really helped. In-person, the person can touch the other person, can reassure the older folks. I think for me that would be really helpful, and they can look right into the person's eyes, online you can, but in-person would be a lot better. (TI 12)*

*b. Challenges with videoconferencing.* Several participants shared their frustrations with technical glitches due to network issues as a result of living in rural areas and winter storms. Some PLwD also required help from their trusted individuals or PI for videoconferencing set up. One PLwD described their experience below:

*The only thing wrong with this whole process was our equipment and our lack of quality of internet. We had one session where her (SV) image froze on the screen and I couldn't get her facial expressions. . . COVID has made it so much harder, the fact that we have to do it over the computer, instead of sitting across from each other has made it so much harder, though it was completely successful. (PLwD 15)*

Overall, the researcher had a positive experience with the videoconferencing technology for ACP discussions. Access to a safe virtual platform broadened the geographic reach for recruiting participants, several of whom lived in rural areas, at about a 200 km distance from the interventionist (PI). Most participants were quite interactive during the sessions–they asked questions, expressed their views, and clarified each other's points too, indicating their comfort with the technology and the interventionist.

That said, there were challenges with delivering the intervention virtually, including environmental factors such as unexpected visitors (e.g., mail delivery) or smoke alarms. In some cases, the trusted individuals would make concerted efforts to have their older adult speak while they were quietly thinking, possibly distracting the PLwD's train of thought. This is likely due to being in a virtual setting, whereas such self-reflection would be more organic and acceptable in an in-person setting. In some cases, where the trusted individuals visited the PLwD for the sessions, both participants wore personal protective gears such as masks, gloves, and gowns due to fear of contracting/transmitting the COVID-19 virus. In one instance, this created additional challenges for the PLwD, requiring them to simultaneously deal with their hearing aid, safety gears, and technology, though they continued participation. Finally, there was some difficulty gauging subtle, non-verbal expressions on the computer screen. To overcome this, participants were frequently asked about their emotions and if they wanted breaks. One participant described her experience:

> She (SV) read my husband very well. He has a speech problem because of dementia. He is not able to communicate very well, but she read his facial expressions very well; she read his tone very well; she understood his non-wording very well. So, I think she did a great job working with a dementia person and the condition that he is in. (TI 18)

On the other hand, one trusted individual felt that their expressions were not picked up on adequately:

> I was just curious to know why it had to be visual, I thought it might be to like observe participants in case there was an intervention that needed to be done, but then I didn't really notice moments of intervention . . . like I know there was one session where I was quite upset and sad and tearful, and that wasn't acknowledged, which is fine; I can process that on my own, like I have my own support networks, but I think in other situations, that might be quite tough for somebody to manage on their own. (TI 9)

Overall, although technological aspects had some impacts on intervention delivery, there were no serious impediments. However, as much as the interventionist attempted to be present with the participants, videoconferencing did impact her ability to pick up on some emotions that may have been diminished due to the virtual platform.

## Discussion

In the current pilot study, VYV intervention was utilized to engage community-dwelling older adults with mild dementia and their trusted individuals in the ACP process. All participants perceived VYV as valuable and acceptable. All PLwD effectively engaged in the ACP process, they shared their values and wishes for future care in the context of advanced dementia and other terminal conditions or being in a vegetative state, which they had not done before. The process also allowed trusted individuals to begin their journey as the decision-maker on behalf of their PLwD as they progress through their disease. The presence of an interventionist with expertise in geriatrics was a critical reason for VYV being considered acceptable. Overall, the

video and sound quality were rated highly by all participants, demonstrating the utility of a virtual modality as a suitable means to conduct ACP discussions for this group without accessibility issues such as blindness or deafness.

Given the small sample size, only the influence of three predictors of interest was examined on the acceptability scores in all participants. Only PLwD living with their trusted individuals had a positive influence on their acceptability of the VYV intervention. This may be due to a concern about becoming a burden for the trusted individuals in the future [35], and making them aware of their future care wishes through VYV may help ease the burden of proxy decision-making. Living with the trusted individual may increase a sense of confidence and pride [35] that may lead to increase comfort in participating in the VYV intervention and a higher level of acceptability. In terms of the other two predictors, sex and timing of dementia diagnosis, no differences were found. Previous research indicates that male and female care partners experience caregiving in a variety of different ways across the trajectory of dementia [36]. Also, males with a dementia diagnosis showed a higher tendency to participate in ACP than females [37].

In terms of timing, it is recommended to commence ACP early in the dementia trajectory before the loss of decision-making capacity [38]. In this study, all older adults had mild dementia, though there was wide variability in the time since dementia was diagnosed. This may hold particular clinical relevance for initiating ACP discussions in persons with mild dementia when they are generally able to effectively express their values and wishes. However, how soon this conversation should be initiated after the diagnosis remains a question that needs further exploration. It is critically important to acknowledge the importance of dementia-specific ACP given the unique and complex trajectory of dementia compared to other chronic illnesses.

As mentioned, only a few intervention studies exist that involved individuals with mild dementia living in community settings in the ACP process [19, 39–41]. Our findings are consistent with these studies that individuals with mild dementia can effectively articulate their values [19, 38]. Our findings also highlight the relationship between tailored education delivered by an interventionist with expertise in the care of older adults and the older adults' willingness to express values and wishes for future care. The VYV intervention was unique as it was delivered to the dyad together, providing trusted individuals the opportunity to not only learn about their PLwD's views for future care, but also seek clarifications. They also got an opportunity to learn more about resources to prepare themselves to continue to be a partner in their PLwD's care, including where to seek help when caring becomes overwhelming, contingency planning in case they get sick or pass before their PLwD, and other available community resources.

Several barriers to dementia-specific ACP have previously been identified by PLwD and their care partners [15]. Similar views were expressed by participants in this study. For example, participants expressed the need to stay in the moment and not think what might come in the future, and many PLwD believed that their trusted individuals already knew about their wishes, which was not necessarily the case. During the VYV sessions, these barriers were addressed through: 1) tailored education about dementia, 2) promoting understanding of the need to engage in the ACP process, and 3) coaching to identify and share values and wishes for future care. The presence of a knowledgeable clinician as the interventionist, with the ability to respond to in-the-moment questions, also facilitated an increase in the participants' self-efficacy to move on in their ACP journey.

As mentioned, the VYV intervention was delivered virtually due to COVID-19. Though there were challenges, the results of this study demonstrate that videoconferencing is a viable tool to engage PLwD and their trusted individuals in dementia-specific ACP discussions.

Participants connected from diverse locations, including rural and urban areas, spread over large geographic distances. This widespread reach would not have been possible had the intervention taken place in-person. That said, all PLwD had their trusted individuals available in-person to provide support with set up and some also required a telephone call from the PI to help connect to MS Teams. This was also important because most were used to other video-conferencing applications such as FaceTime or Zoom. As such, future research and clinical work involving videoconferencing should consider allowing participants to use the platform they are most comfortable with for greater buy-in, and to reduce the stress associated with the initial connection [42].

Study participants were provided some basic equipment such as headphones, splitters, and dongles. These helped optimize participants' hearing experience, especially for those who had mild hearing loss. However, much of the success of initiatives involving videoconferencing requires a high quality connection of 1024 kbps bandwidth [17]. Some of the participants who lived in the rural regions experienced choppy signals at times, which impacted the overall experience. Future studies may consider providing participants with mobile internet devices for better connection experience. Other researchers have successfully used secure telehealth network housed in a local hospital or clinic where participants could easily travel for interventions such as support group and memory assessment [42, 43]. As it is, persons with dementia are marginalized when it comes to ACP due to barriers associated with a decline in cognitive capacity [44]. Providing a virtual option for the delivery of ACP discussions is beneficial to broaden access to these important services, especially in a geographically expansive country like Canada. Post-pandemic, it will also be important to continue to provide an in-person option for PLwD who are not comfortable with technology, those who require accessibility accommodations, or those with a greater preference for in-person discussions.

The study had some limitations. Firstly, the TEI acceptability questionnaire had not previously been tested in remote research, although internal consistency was acceptable. To ensure acceptability of the VYV intervention, data from open-ended questions and the interventionist diary log were used to better interpret the quantitative data to better understand participants' views. The ultimate hope is that ACP is embedded in the routine dementia care, so PLwD and their trusted individuals are less apprehensive than if discussions linked to poor prognosis or lack of treatment options. The findings of this study are promising to support the appropriateness of ACP in early dementia. The VYV intervention can be scaled up through measures to upskill the health care professionals including those in the primary care practices through education and resource allocation. One of the components of the VYV intervention is education to the participants and the ability to respond to questions that arise in the moment. Therefore, any clinician who has training in the care of older adults and how to broach the ACP conversation is important. One of the possible barriers may be related to time constraints in these practices and therefore, implementing interdisciplinary models where realization of ACP initiatives may be more tenable. Future studies should also enroll larger sample sizes to further establish the reliability of the above measures and employ both quantitative and qualitative data to triangulate findings [45]. Furthermore, long-term data was not collected to determine if expressed values and wishes translated into care received at the end-of-life. As such, there is potential for a future longitudinal study to conduct such analysis.

## Conclusion

This research contributes to the ACP discourse in multiple ways. Firstly, despite its demonstrated benefits, ACP for those with mild dementia remains uncommon. The VYV study provides a model that could facilitate these important discussions. Secondly, findings from this

study build on a growing body of research illustrating the need to incorporate videoconferencing as a viable and acceptable option to deliver a tailored ACP intervention to PLwD and their trusted individuals in community settings.

## Supporting information

**S1 Table. Categories and subcategories related to the acceptability of VYV with additional quotes.**
(DOCX)

**S1 File.**
(DOCX)

## Acknowledgments

We would like to thank all the individuals with dementia and their trusted individuals who were gracious with their time; and for their efforts to participate in this important project. This research was supported by Maria and Walter Schroeder Institute for Brain Innovation and Recovery.

## Author Contributions

**Conceptualization:** Shirin Vellani.

**Data curation:** Shirin Vellani.

**Formal analysis:** Shirin Vellani.

**Investigation:** Shirin Vellani.

**Methodology:** Shirin Vellani.

**Project administration:** Shirin Vellani.

**Software:** Shirin Vellani.

**Supervision:** Martine Puts, Andrea Iaboni, Katherine S. McGilton.

**Validation:** Shirin Vellani.

**Visualization:** Shirin Vellani.

**Writing – original draft:** Shirin Vellani.

**Writing – review & editing:** Shirin Vellani, Martine Puts, Andrea Iaboni, Katherine S. McGilton.

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
