## [Decision Letter · Decision Letter 0]

29 Nov 2021

PONE-D-21-24591Acceptability of the Voice Your Values, a tailored advance care planning intervention in persons living with mild dementia using videoconferencing technologyPLOS ONE

Dear Dr. Vellani,

Thank you for submitting your manuscript to PLOS ONE. After careful consideration, we feel that it has merit but does not fully meet PLOS ONE’s publication criteria as it currently stands. Therefore, we invite you to submit a revised version of the manuscript that addresses the points raised during the review process.

Please address the reviewers' comments and concerns, especially those related to the reproducibility and adequate reporting of the study. 

We look forward to receiving your revised manuscript.

Kind regards,

Lucy Selman, PhD

Academic Editor

PLOS ONE

Journal Requirements:

Reviewers' comments:

Reviewer's Responses to Questions

**Comments to the Author**

1. Is the manuscript technically sound, and do the data support the conclusions?

Reviewer #1: Partly

Reviewer #2: Yes

2. Has the statistical analysis been performed appropriately and rigorously? 

Reviewer #1: Yes

Reviewer #2: Yes

3. Have the authors made all data underlying the findings in their manuscript fully available?

Reviewer #1: No

Reviewer #2: Yes

4. Is the manuscript presented in an intelligible fashion and written in standard English?

Reviewer #1: Yes

Reviewer #2: Yes

5. Review Comments to the Author

Reviewer #1: Review ”Acceptability of the Voice Your Values, a tailored advance care planning intervention in persons living with mild dementia using videoconferencing technology”

Thank you for the opportunity to review this manuscript on an interesting and important topic. My impression is that the authors want very much with this manuscript. There are several research questions posed but all of them are not answered in a satisfying way. The methods section also lacks important information needed to understand the intervention in full. The manuscript has potential to be a nice article but it needs some work in order for that.

Abstract

At one place in the abstract, line 18, Voice Your Values has been given VVY as abbreviation.

Introduction

The authors describe ACP and research about that in a short summative way and puts the study in the time and place of the covid-19 pandemic.

Method

Important details of the method is missing, see comments below, and for that cause the study is not possible to replicate.

Design

In the design section you state that the feasibility of the intervention is reported on elsewhere, and give the reference for that study. That is good, but could you please give one short concluding statement about the feasibility in this paper. Not all readers will bother to search for the referenced article. I presume that the VYV was found feasible otherwise I guess you would not have proceeded with present article.

Participants

I wonder if not the interventionist also should be mentioned here? This person is also a kind of participant. I am aware that you write about the interventionist at line 53 but I see that this information rather belongs here since the interventionist participates in the intervention and also evaluates the intervention.

Voice Your Values intervention

In line 104 a researcher named Piers is referenced to for recommendations about how to deliver the intervention but no reference is given for this. I also find no Piers at all in the rest of the manuscript.

Were the interventionist/s working at any of the five participating geriatric clinics? Did the interventionist have any prior relationship to any of the participants through hir work?

Measures

It is not fully clearly stated how the research assistant collected the data. Where this done in a virtual meeting? How was data captured, written down or by audio-recordings?

Data analysis

In this section the interventionist’s diary log is stated to be used for analysis. Besides from the abstract and at one place in the discussion, is this the only place where the diary log is mentioned. Please add information about the diary log, what was written in it and when under the appropriate heading above in the manuscript (in the methods section).

Ethical considerations

It is not clearly stated which relation the interventionist had to the participating persons living with dementia. It is neither stated if the intervention was given as part of a new way of working within the five study sites or if the intervention was only performed as part of the research project. Information about the research assistant’s relations to the participating dyads is neither stated. Did this person have a caring relation to the participating person living with dementia or was the research assistant only part of the research group? These issues are important to make clear to understand if there were any power relations to be aware of between the researchers (the interventionist and the research assistant?) and the participants.

Results

The study states to have three research questions:

1. Acceptability of the intervention

2. Factors influencing the acceptability

3. Experiences with video conferencing

As I understand, the quantitative analysis answers the first and the second question but the manuscript states that acceptability is answered through the qualitative analysis. However, I am not convinced about that. TEI is stated to measure acceptability and TEI scores are used for the quantitative analysis. I understand that you had added six open ended questions to the TEI, and as far as I understand the answers to these questions are used in the qualitative analysis. So one part of the acceptability might be answered by your qualitative analysis but one part is answered by the quantitative analysis. Please adjust the text about which of the research question the different parts of the results are answering.

The quantitative findings are pretty straightforward reported. However I struggle with the qualitative findings. Firstly one question about terminology. In table 4 you state that “Breaking the ice” and all the other “categories” are subcategories but in the text below the table you write about them as categories. At one place you also use the word theme for the results of the qualitative analysis. Please revise for clarity and consistency.

I am also not convinced that the five categories answer any of the questions of the study. I have problems to interpret the five categories as describing “acceptability of the intervention”. Some of them might be possible to understand as a description of acceptability, but “I want to die on my terms” is very hard to interpret as if it deals with acceptability of VYV.

The second category is also formulated so that only the participating persons living with dementias statements can be included. I would suggest to change the category name in a way that the trusted individual’s views can be included too.

The qualitative analysis needs to be remade with the research question about “acceptability of the intervention” kept in mind. One other way to sort this out might be to change the research question to fit the analysis that has been made. Maybe the last of the research questions could be rephrased as two? One about experiences of videoconferencing for VYV (which is already answered) and one about what help VYV provides in talking about how one wants to be cared for the last time in life and how one wants to die, or something similar?

For the last part of the results, the part reporting on participants experiences of engaging in VYV virtually is in table 4 stated to be constituted of two sub categories. The categories can be found in the text but in line with how the other equal parts of the results are presented I suggest that the sub categories names are used as headings.

Discussion

In the discussion reasons for why VYV was considered acceptable comes forth (lines 389-392). This is the kind of result that I would have expected if acceptability was in focus but the result is not reported in this way.

The discussion also brings forth findings from present study that I do not recognise from the results such as barriers to ACP (lines 425-429). If barriers found through the analysis are going to be discussed they first need to be reported on in the results section.

Strength and limitations

Some limitations of the study are discussed but not the fact that the intervention was performed by the PI of the study. In a future and if the intervention is going to be used as part of daily clinical routines for persons living with dementia the intervention will need to be performed by the regular health care professionals at the geriatric clinics. This fact needs to be discussed not least since the PIs good skills were lifted as a key factor for the acceptability of the intervention. What are your thoughts of the possibility of scaling up the intervention? What potential barriers and possibilities do you see? Do you think that an education will be needed for health professionals to be able to be in charge of the VYV intervention?

References

Titles of cited articles are alternately written with sentence style and with headline style and the name of the journal is likewise written out in full, with and without capitalisation and written in their abbreviated form. Please adhere to the author guidelines for these two issues.

I would suggest that you consider changing reference 33 to another one. There must be a written references where this can be found.

Table 1

In the list of common chronic conditions you have listed MSK conditions as well as HTN without giving the abbreviated expression while you are not abbreviating coronary artery disease with CAD. In the name of logic, this would have been what I had expected. I would suggest that you write the expressions out in full since I am pretty sure that these abbreviations not are fully transparent for all potential readers.

I am also surprised to not find the very common cerebrovascular diseases among the chosen chronic conditions, since it is one of the most common chronic conditions globally and not least since it also is a common contributing underlying cause to dementia.

Why is there no age range for the trusted individuals? If a mean with SD have been calculated, the range must be known.

Table 2

It would be good if min and max score for the participants experiences were described in proximity to the table itself or where the questionnaires are described.

Table 3

Table 3 gives the impression that you have calculated the time since the trusted individuals were diagnosed with dementia. Is that what have been done? Maybe I read the table the wrong way?

Table 4

I have difficulties understanding this table and there are several issues:

The table text states that categories and themes are given in the table. However the word theme is not present anywhere in the table.

I the first column it I stated that five categories were established but I struggle to find them. Is Acceptability one category? And is Participants’ experiences of engaging in VYV virtually? In that case I find two categories.

I also have difficulties with understanding the last column with Description of the subcategories. How is “VYV as a means to engage in ACP process” a description of the subcategory “Breaking the ice”? The same applies for all description in this column.

Table S1

It is a bit difficult to understand which of the quotes belongs to which subcategory since no vertical lines that guides the reader, or extra free space is present. The quotes adds no value if the reader will have to guess which category they illustrate.

Language

Line 34 -35 contains an “any” each, that is two in a row in that sentence.

Deposition of data

In the part of the submission that I have got access to it is stated that all data are fully available without restriction but I cannot find any information about where the data are deposited.

Guidelines for studies

No guidelines for publication is referred to in the manuscript and no guideline document is included in the part of the submission that I have access to.

Reviewer #2: thanks for this interesting study, i enjoyed reading this article. i dont have many comments other than one every minor below:

The details of the person who facilitated the intervention should be in the intervention section rather than in the introduction.

In the discussion more could be said about future research - what is next for this study? I don't think a trial is needed but an implementation study would be good to see.

What are the implications for other conditions?

Does the facilitator need to be a geriatrician or nurse - could it be someone else? This would be an expensive model so what are the alternatives or is it about implementing it into routine practice?

6. PLOS authors have the option to publish the peer review history of their article (what does this mean?). If published, this will include your full peer review and any attached files.

Reviewer #1: No

Reviewer #2: No

---

## [Author Response · Author response to Decision Letter 0]

28 Dec 2021

Detailed response letter with itemized responses are attached as a file.

---

## [Editor Report · Decision Letter 1]

29 Mar 2022

Acceptability of the Voice Your Values, an advance care planning intervention in persons living with mild dementia using videoconferencing technology

PONE-D-21-24591R1

Dear Dr. Vellani,

We’re pleased to inform you that your manuscript has been judged scientifically suitable for publication and will be formally accepted for publication once it meets all outstanding technical requirements.

Kind regards,

Lucy Selman, PhD

Academic Editor

PLOS ONE
---

## [Editor Report · Acceptance letter]

6 Apr 2022

PONE-D-21-24591R1 

Acceptability of the Voice Your Values, an Advance Care Planning Intervention in Persons Living with Mild Dementia Using Videoconferencing Technology 

Dear Dr. Vellani:

I'm pleased to inform you that your manuscript has been deemed suitable for publication in PLOS ONE. Congratulations! Your manuscript is now with our production department. 

Kind regards, 

on behalf of

Dr Lucy Selman 

Academic Editor

PLOS ONE